# Factors Influencing Suicidal Ideation in Persons with Physical Disabilities

**DOI:** 10.3390/bs14100966

**Published:** 2024-10-18

**Authors:** Eun-Young Park

**Affiliations:** Department of Secondary Special Education, Jeonju University, Jeonju 55069, Republic of Korea; eunyoung@jj.ac.kr

**Keywords:** suicidal ideation, physical disabilities, sex, age, depression

## Abstract

Suicidal ideation is a leading indicator of suicide risk, particularly among persons with disabilities, a population at high risk of suicide. This study aimed to identify factors affecting suicidal ideation among persons with physical disabilities in Korea. Data for this cross-sectional study were obtained from the Disability and Life Dynamics Panel. Eight hundred and twenty-eight persons with physical disabilities were extracted from the data, and seven hundred seventy people who responded to the suicidal ideation question were included in the analysis. Chi-square and multivariate logistic regression analyses were employed. The results show a high percentage of suicidal ideation among persons with physical disabilities (18.5%). Among the general characteristics, more women were found to have suicidal ideation than men, and people in their 40s and 60s had a higher rate of suicidal ideation than other age groups. A lower educational level was found to be related to suicidal ideation. After controlling for general characteristics, depression increased the likelihood of suicidal ideation among individuals with physical disabilities. Economic difficulties and disability acceptance did not relate to suicidal ideation. Based on the results of this study, continuous observation of suicidal ideation in people with physical disabilities and early intervention programs for preventing depression and increasing disability acceptance are considered necessary.

## 1. Introduction

Despite significant global prevention efforts, suicide remains a critical health issue because of its substantial impact on global mortality rates [1,2]. As of 2020, the number of people who died from intentional self-harm, or suicide, was 26.9 per 100,000 people, and Korea still has the highest suicide rate among OECD countries [3]. Suicide rates are even higher among people with disabilities. A study analyzed data from the 2015 to 2019 National Survey on Drug Use and Health, focusing on 198,640 adults in the U.S., and found that 19.8% of participants reported having a disability and that individuals with disabilities were significantly more likely to report suicidal ideation, planning, and attempts than those without disabilities [4].

Although most people agree that suicide is a serious social problem, empirical data on whether its risk among people with disabilities is higher than among those without disabilities is lacking. A 2018 social survey revealed that economic difficulties, diseases, and disabilities were the leading causes of suicidal ideation and that physical disabilities increase suicidal ideation [5,6]. The presence of physical disabilities is linked to mental health issues like depression, anxiety, and post-traumatic stress disorder, all of which increase the suicide risk [7]. Research on suicidal ideation among individuals with physical disabilities in the Korean context is crucial due to the unique cultural and social factors that may influence mental health outcomes. In South Korea, societal pressures, the stigma surrounding disability, and the high value placed on self-sufficiency [8,9] can exacerbate feelings of burdensomeness and isolation among people with physical disabilities. Perceived burdensomeness, a key concept in the Interpersonal Theory of Suicide, is the belief that one is a burden to others to the extent that others might be better off if they were dead [7]. Research shows that movement disorders, a type of physical disability, are linked to feelings of being a burden and depression in adults [10].

A Canadian study found that the risk of suicide among people with disabilities was higher than among those without disabilities [11]. In addition, Giannini and Bergmark [12] compared the suicide risk of people with and without disabilities based on the results of 21 studies on suicide in three disability types (spinal cord disability, intellectual disability, and multiple sclerosis disorder) and found that the suicide risk of people with disabilities was higher. However, previous studies that reported high suicide rates among people with disabilities also point out a lack of data on suicide among people with disabilities and argue for the need for future research on suicide risk among people with disabilities.

People with physical disabilities (PWPD) bear a large financial burden due to continuous treatment and rehabilitation and incur additional expenses for home adaptations, social care support, and mobility or communication aids owing to their limitations [13]. They are prone to psychological decline and depression due to their physical disabilities [14]. Russell and Turner [5] report that this is more than people without disabilities. Awareness and discussion of suicide by PWPD increases social interest in the problem of suicide among people without disabilities.

Damage caused by a disability can decrease physical and psychological efficacy and cause mental stress because it limits certain activities or independence in daily life [15]. Overall, the traumatic event of developing a physical disability affects households with persons with disabilities as a whole. People with disabilities experience psychological difficulties, such as stigma and post-traumatic stress [16]. Unlike other types of disabilities, physical disabilities often occur because of suddenly acquired reasons, such as accidents [17]. In such cases, in addition to physical activity limitations that PWPD had never experienced before, they also experience social prejudice due to their externally visible disabilities [18]. This makes it difficult for them to adapt to and accept their physical and mental damage, and they experience various psychological maladjustments as they feel anxiety and anger at their appearance and reality, which have changed since before becoming disabled. This can cause suicidal ideation [19,20], and can lead to suicide; therefore, it should be regarded as an important signal [21]. To reduce the suicide rate of people with disabilities, it is necessary to analyze the causes and risk factors of suicide among people with disabilities in depth and establish systems and policies to prevent suicide [6]. Given that each type of disability has unique characteristics, developing an intervention program based solely on general information about persons with disabilities, without specifying the type of disability, is challenging. This study aims to address this gap by focusing specifically on PWPD, offering detailed insights into their unique needs. Therefore, this study aimed to identify the factors related to suicidal ideation in PWPD, who have a high rate of suicidal ideation.

The specific research questions were followed. First, were there differences in suicidal ideation according to the general characteristics of PWPD? Second, which factors influenced suicidal ideation in PWPD?

## 2. Materials and Methods

### 2.1. Study Design

This study employed a cross-sectional survey design.

### 2.2. Setting

This study utilized the first (2018) data from the Disability and Life Dynamics Panel (DLDP). The DLDP is a survey conducted by the Korea Disabled People’s Development Institute since 2018, constructing a panel of 6121 persons with disabilities nationwide. The purpose of the DLDP is to identify the dynamics of personal, family, and social factors regarding the process of disability acceptance and changes experienced in social relationships due to the occurrence of disability and to secure basic data necessary for establishing and supporting related policies in the future. The DLDP is a nationally approved statistic implemented under the Statistics Act of South Korea.

### 2.3. Participants

The inclusion criteria were registered PWPD aged 18 years or older residing in the Republic of Korea. The exclusion criterion was a failure to respond to the suicidal ideation question. Table 1 shows the characteristics of PWPD in this study. The participants were slightly more male (51.8%) than female (48.2%). The majority of individuals are aged between 50 and 69 years, with the largest group in the 60–69 age range (34.9%). Most participants have a high school education (40.5%), with a significant number having completed middle school (20.8%) or elementary school (20.0%). The average depression level was 10.73. When converted to a score of 20 items, the percentage of people who scored 16 or higher, which is the standard for being depressed, was found to be 59.8%.

### 2.4. Outcome Measures

The dependent variable, suicidal ideation, was measured by responses to the question, “Have you ever thought deeply about suicide?” with yes (coded as 1) and no (coded as 0). General characteristics included gender, age group (10–19, 20–29, 30–39, 40–49, 50–59, 60–69, 70 or more), and the highest level of education (no education, elementary school, middle school, high school, above college). Depression was measured using the 11-item Center for Epidemiological Studies Depression Scale, with responses measured on a 4-point Likert scale; higher scores indicate more severe depressive symptoms. The reliability of the scale was 0.817 (0.798–0.835) in this study. Previous studies have confirmed the reliability and validity of the CES-D 11 for PWPD [14]. Disability acceptance was measured using 12 items developed by Kaiser and Wingate [22] which were revised and supplemented. Examples of items include “I have a disability, but I am satisfied with my life” and “I am not distressed by my disability”. A higher score indicates a higher level of acceptance of disability. The reliability was 0.754 (0.729–0.778) in this study. Economic hardship consisted of eight items such as I have difficulty forming relationships because of money, and I do not have the financial means to enjoy culture and leisure activities. Each item was measured from 1 (not at all) to 4 points (very much), with a higher score indicating greater economic hardship. The reliability was 0.912 (0.902~0.921) in this study.

### 2.5. Bias

We attempted to minimize the following biases. To avoid selection bias, the sample in this study was distributed and extracted by considering the type of disability, degree of disability, and gender, according to the results of the sample design research, to increase the degree of representation and estimation of the population. Multivariate logistic regression analysis was used to control for confounding variables such as age, sex, and educational level. The reliability of responses and the minimization of measurement bias were ensured by having non-researchers conduct the survey through face-to-face interviews.

### 2.6. Sampling and Sample Size

The number of participants in the DLDP in 2018 was 6512, and data from 828 PWPD were extracted for analysis. Data from 770 people who responded to the suicidal ideation questionnaire were finally analyzed. To correct sample bias, weights were used for analysis.

### 2.7. Statistical Analysis

Frequency analysis was conducted using SPSS 26.0 to describe the characteristics of the PWPD. Differences in suicidal ideation among PWPD according to demographic characteristics were assessed by the chi-square test (*X*^2^). Specifically, in the post-hoc test for *X*^2^, we analyzed the adjusted standardized residuals with Bonferroni correction, which is an adjustment made to the *p*-values when several dependent or independent statistical tests are performed simultaneously on a single dataset. Pearson’s correlations were used to detect multicollinearity among the variables, and there were no multicollinearity problems. Multivariate logistic regression analysis was used to examine the relationships between sex, age, educational level, depression, economic difficulty, disability acceptance, and the presence or absence of suicidal ideation. This study utilizes a secondary analysis of previously collected and publicly accessible data. As a result, ethics approval was not necessary according to relevant institutional and national guidelines and regulations.

## 3. Results

The results of the examination of the differences in suicidal ideation according to demographic characteristics are shown in Table 1. There were differences in suicidal ideation according to gender, age, and educational level. Females with a physical disability had significantly more suicidal ideation than men with a physical disability. In total, the rate of suicidal thoughts among PWPD was 18.5%. The results of the post-test showed that PWPD in their 40s and 60s had significantly more suicidal ideation than those in other age groups. The rate of suicidal ideation was significantly higher in the group of elementary school graduates and no education, while the rate of suicidal ideation was significantly lower in the group of high school and above college graduates.

The results of the logistic regression analysis used to identify the variables related to suicidal ideation are shown in Table 2. Among gender, age, educational level, depression, economic difficulty, and disability acceptance, gender, age, depression, and disability acceptance were associated with suicidal ideation. When depression increased by 1, the possibility of suicidal ideation occurring increased 1.174 times. When disability acceptance increased by 1, the possibility of suicidal ideation occurring decreased by 0.981 times.

## 4. Discussion

This study investigated factors affecting suicidal ideation among persons with physical difficulties in Korea. Not all suicidal ideation leads to suicidal acts, but it is a process of suicidal behavior that leads to suicide attempts and actual suicide [23], and even mild suicidal ideation can later develop to a serious level or lead to suicide. Hence, suicide ideation should be considered an important warning message [24].

In this study, the percentage of PWPD with suicidal ideation was 18.5%, which was much higher than the worldwide lifetime prevalence of approximately 9.2% [25]. The logistic regression showed that gender, age, depression, and disability acceptance have been related to suicidal ideation in this study. The results of this study showed that women were significantly more likely to have suicidal ideation than men, which is not consistent with the results of a previous study [26]. Research on depression consistently shows that women are more likely than men to experience depression [n]. Given that depression is significantly associated with suicidal ideation, the findings of this study, which show that women are more likely than men to have suicidal thoughts, align with this broader context. This suggests that the higher prevalence of depression among women may contribute to their increased likelihood of experiencing suicidal ideation compared to men. In terms of age, the incidence of suicidal ideation was significantly higher among those in their 40s and 60s. A UK study confirmed that the 16–34 age group was 3.64 times higher in suicidal ideation than the 55–74 age group, and the 35–54 age group was 2.71 times higher than the 55–74 age group [27]. In addition, the rate of suicidal death was higher in the low-education group with a high school diploma or lower, and education was reported to be a major factor in predicting suicide deaths [28]. However, the results of this study showed that educational level had no significant effect on suicidal ideation.

When factors affecting suicide among people with disabilities were analyzed, and demographic variables were controlled in logistic regression analysis, depression and disability acceptance were significantly related to suicidal ideation. Among the general population, factors related to suicide include a family history of psychiatric illness, depressive mood, excessive anger, and inadequate or excessive sleep [29]. In addition, in the case of people with disabilities, the higher the stress [5] and the lower the acceptance of disability [30], the higher the likelihood of suicidal ideation. Depression can be said to be a psychological characteristic that an individual face and this psychological state is a decisive factor in causing suicide [31]. In fact, previous studies have shown that the suicide rate of people with disabilities is higher than that of people without disabilities, and the risk increases when they have psychological problems such as depression or anxiety [19]. In a recent systematic review, Lutz and Fiske [32] found that adolescents with physical disabilities were significantly more likely to commit suicide or have suicidal behavior than adolescents without physical disabilities and that depression plays a role in the relationship between disability and suicide-related outcomes. Disability acceptance involves recognizing and embracing one’s disability without devaluing oneself, acknowledging physical limitations, and enhancing one’s sense of worth despite the disability [33,34]. Those who successfully accept their disability tend to have a higher quality of life, experience positive growth, and benefit in areas such as employment and social reintegration [35]. This process fosters a positive self-concept, emotional stability, and social connectedness. As the final stage of psychological adjustment to disability, acceptance allows individuals to reevaluate their life goals and pursue new meanings, which can help maintain quality of life and reduce suicidal ideation in those who acquire a disability later in life.

According to Harris and Barraclough [36], people who had suicidal ideation were 47 times more likely to end their lives by suicide than those who did not. Suicidal ideation usually leads to a suicide attempt within one year; 34% of those who had suicidal ideation proceeded to the stage of planning suicide, and 72% attempted suicide [37]. A meta-analysis by Franklin and Ribeiro [38] found that the primary risk factor for future suicidal ideation was prior suicidal ideation, emphasizing its recurrent and chronic nature. Suicidal ideation was the third most potent predictor of death by suicide, following prior psychiatric hospitalizations and prior suicide attempts.

It would be desirable to study suicidal behaviors or cases directly; however, since epidemiological studies are difficult in reality, suicidal ideation or suicide attempts have been used as important indicators and proxy variables for suicide. Suicidal ideation is an important indicator for preventing or understanding the risks that may lead to suicidal behavior; therefore, identifying the factors of suicidal ideation is of great significance. This study explored the factors affecting suicidal ideation in PWPD and found that there was a relationship between gender and age among general characteristics; when general characteristics were controlled for, it was found that depression could significantly increase suicidal ideation. The percentage of PWPD with suicidal ideation was high. Considering that suicidal ideation is a leading indicator of suicide risk, continuous observation of suicidal ideation in PWPD and early intervention for PWPD with suicidal ideation are necessary. In addition, as in other population groups, depression was an important predictor of suicidal ideation in PWPD. Therefore, to prevent suicidal ideation in persons with disabilities, intervention programs for depression prevention, early detection, and treatment appropriate for the situation and needs of this population should be provided.

The significance of this study lies in the fact that it analyzed variables related to suicidal ideation in PWPD using a large sample and systematic sampling methods. Nevertheless, this study has limitations that can appear in a panel data analysis. Although there may be variables related to suicidal ideation other than those analyzed in this study, only the data presented in the panel could be analyzed. In future studies, research on variables such as stress and social support, which have been reported as variables affecting suicidal ideation, is recommended. Secondly, this study analyzed panel data with no sample attrition to ensure sample representativeness and thus utilized data collected before the pandemic. Because the pandemic has had such a significant impact on our lives, research is needed on how the pandemic has affected suicidal thoughts in people with disabilities. The findings from this study, which analyzed pre-pandemic data on thoughts and behaviors of individuals with a thumb disability, suggest that understanding pre-existing tendencies is crucial for distinguishing patterns and studying emotional responses during and after COVID-19. Thirdly, suicidal thoughts were measured dichotomously. Because dichotomous measures may not reflect the complexity of suicidal ideation, future research is needed to reflect various aspects of suicidal ideation.

The results of this study, which investigated factors related to suicidal ideation in people with physical disabilities, can be used as data for future customized interventions. In particular, the results of this study suggest that policies and support services for women, those in their 40s and 60s, and those with physical disabilities with low levels of education should be considered among general characteristics. In order to reduce suicidal ideation, mental health intervention programs should be focused on reducing depression and should be improved to increase disability acceptance.

## 5. Conclusions

The findings showed a high percentage of suicidal ideation among PWPD (18.5%). Additionally, more women were found to have suicidal ideation than men, and people in their 40s and 60s had a higher rate of suicidal ideation than other age groups. The rate of suicidal ideation was significantly lower in the group of high school and above college graduates than in elementary school graduates and those with no education.

Logistic regression results showed that gender and age have a significant relation with suicide ideation. After controlling for general characteristics, depression increased the likelihood of suicidal ideation, and disability acceptance decreased the likelihood of suicidal ideation. Continuous observation of suicidal ideation in PWPD and early intervention for those with suicidal ideation are recommended based on the findings. In addition, intervention programs for depression prevention and increasing disability acceptance, early detection, and treatment appropriate for the situation and needs of this population should be a priority.

## Figures and Tables

**Table 1 behavsci-14-00966-t001:** Differences in suicide ideation according to demographic characteristics.

Categories	Total		Suicide Ideation		No Suicide Ideation		
	n	%	n	%	n	%	*X* ^2^
Sex							
Male	23,933	56.7	3739	48.1	20,194	58.6	284.518 **
Female	18,032	43.3	4034	51.9	14,268	41.4	
Total	41,965	100	7773	18.5	34,462	82.1	16,863.36 **
Age							
10–19	889	2.1	37	0.5	852	2.5	361.800 **
ASR			11.1 **		−11.1 **		
20–29	491	1.2	94	1.2	397	1.2	
ASR			−0.4		0.4		
30–39	1062	2.5	132	1.7	930	2.7	
ASR			5.1 **		−5.1 **		
40–49	2510	5.9	658	8.5	1852	5.4	
ASR			−10.4		10.4		
50–59	14,409	34.1	2664	34.3	11,745	34.1	
ASR			−0.3		0.3		
60–69	17,794	42.1	3518	45.3	14,276	41.4	
ASR			−6.2 **		6.2 **		
70 or more	5081	12.0	670	8.6	4411	12.8	
ASR			10.2 **		−10.2 **		
Educational level							
No education	1950	4.6	520	6.7	1430	4.1	134.709 **
ASR			−9.6 **		9.6 **		
Elementary school	10,587	25.1	2101	5.0	8486	24.6	
ASR			−4.4 **		4.4 **		
Middle school	8802	20.8	1603	20.6	7199	20.9	
ASR			0.5		−0.5		
High school	15,806	37.4	2725	35.1	13,081	38.0	
ASR			4.8 **		−4.8 **		
Above College	5090	12.1	823	10.6	4267	12.4	
ASR			4.4 **		−4.4 **		

Note: ASR = Adjusted standardized residuals, ** *p* < 0.01.

**Table 2 behavsci-14-00966-t002:** Summary of logistic regression analysis for suicide ideation.

Variable	B	SE	OR	−95% CI	+95% CI	*p*
Gender	0.158	0.027	1.171	1.110	1.981	0.000
Age	−0.076	0.013	0.927	0.903	2.347	0.000
Educational level	0.005	0.009	1.005	0.988	1.423	0.563
Depression	0.160	0.003	1.174	1.167	1.224	0.000
Economic difficulty	0.001	0.003	1.001	0.996	1.061	0.612
Disability acceptance	−0.010	0.005	0.990	0.981	1.072	0.026

Note: B = unstandardized coefficient; SE = standard error; OR = odds ratio; CI = confidence interval.

## Data Availability

You can obtain the data by requesting them from the following site: https://koddi.or.kr/stat/html/user/pwpn/daap/app/pwdlPnlAplyPrcd.do (accessed on 20 September 2022).

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
