# Peer review of "Factors Influencing Suicidal Ideation in Persons with Physical Disabilities"

_behavsci, 2024, doi:10.3390/bs14100966_

Round 1
Reviewer 1 Report
Comments and Suggestions for Authors
Factors Influencing Suicidal Ideation in Persons with Physical Disabilities
This is an interesting and important contribution to the field of mental health of people with physical disabilities. Overall, the article is well written and scientifically structured, but some changes would improve its chances of being published:
1. Abstract: please use a more structured format, including background/methods (participants, measurements, procedures), results and conclusions.
2. Introduction: more references regarding the psychosocial impacts of physical disabilities are needed, focusing on suicidal behavior and ideation in particular.
3. Authors must provide a clear picture of the Korean context, why this study is relevant in this country.
4. Please clearly state the objectives of this study.
5. Data collection took place 6 years ago. Have things changed since then? After the Covid-19 pandemic? Please elaborate.
6. Specific information on the physical disabilities participants had must be provided.
7. Measuring suicidal ideation dichotomously constitutes an important limitation. Please elaborate.
8. What are the depression levels across the sample?
9. Limitations and implications of this study must be discussed.
Best wishes.
Author Response
Reviewer 1:
Summary: This is an interesting and important contribution to the field of mental health of people with physical disabilities. Overall, the article is well written and scientifically structured, but some changes would improve its chances of being published:
Response to summary: Thank you very much for taking the time to review manuscripts. Please find the detailed responses below. The corresponding revisions highlighted in blue where I have made the major changes; please, note that we removed track changes that show all our edits.
Comment #1: Abstract: please use a more structured format, including background/methods (participants, measurements, procedures), results and conclusions.
Response #1: Based on the reviewers' comments, the abstract was revised into a structured format (page 1, abstract).
Comment #2: Introduction: more references regarding the psychosocial impacts of physical disabilities are needed, focusing on suicidal behavior and ideation in particular.
Response #2: The related references based on the comments have been inserted (page 2, line 46~line 50).
Comment #3: Authors must provide a clear picture of the Korean context, why this study is relevant in this country.
Response #3: The related references based on the comments have been inserted (page 2, line 41 ~ 46).
Comment #4: Please clearly state the objectives of this study.
Response #4: Specific research questions have been stated (page 2, line 87 ~ line 89).
Comment #5: Data collection took place 6 years ago. Have things changed since then? After the Covid-19 pandemic? Please elaborate.
Response #5: To ensure the representativeness of the sample, data without sample attrition was used. The need to analyze post-pandemic data is discussed in the discussion section (page 7, line 254 ~ line 261).
Comment #6: Specific information on the physical disabilities participants had must be provided.
Response #6: Information have been inserted data section (page 3, line 108 ~ line 115).
Comment #7: Measuring suicidal ideation dichotomously constitutes an important limitation. Please elaborate.
Response #7: Regarding your comment, limitation have been stated (page 7, line 261 ~ line 264).
Comment #8: What are the depression levels across the sample?
Response #8: The information about depression level have been provided (page 3, line 113 ~ line 116).
Comment #9: Limitations and implications of this study must be discussed.
Response #9: Limitations and implications of this study have been extended (page 7, line 256 ~ line 272; page 7, line 285 ~ line 287).
Reviewer 2 Report
Comments and Suggestions for Authors
Thank you for the opportunity to review this manuscript on an important topic. In general, the paper could use significant editing to reduce redundancy in the text. I hope the following specific comments will help the authors improve their work and its dissemination.
Abstract
· Lines 14 – 16: This statement is an incorrect interpretation of the results in Table 2.
Introduction
· Lines 24 – 28: The incorrect citation is given here. See doi: 10.1016/j.amepre.2021.05.035
Materials and Methods
· Survey data analysis procedures were not described. I would anticipate that the DLDP applies sampling weights. If so, the data analyses and tables need to be corrected.
Results
· Table 1: Please adjust the column label for the “No suicide ideation” group to align with the “n” column for that group.
· Lines 137 – 138: The outcome variable for suicidal ideation was binary, while this statement is expressed as a count variable. Please correct or clarify.
Discussion
· Line 149: The percentage of PWPD with suicidal ideation (18.7%) is not easily discernable from Table 1, nor is it given in the Results section. Please make adjustments to make this clearer.
· Suggest primarily focusing on the adjusted results from Table 2 for the Discussion section.
· Lines 170 – 173: Like the abstract, this statement is an incorrect interpretation of the results in Table 2.
Conclusions
Lines 212 – 214: As previously described, this statement is an incorrect interpretation of the results in Table 2.
Comments on the Quality of English LanguageIn general, the paper could use significant editing to reduce redundancy in the text.
Author Response
Reviewer 2:
Summary: Thank you for the opportunity to review this manuscript on an important topic. In general, the paper could use significant editing to reduce redundancy in the text. I hope the following specific comments will help the authors improve their work and its dissemination.
Response to summary: Thank you very much for taking the time to review manuscripts. Please find the detailed responses below. The corresponding revisions highlighted in blue where I have made the major changes; please, note that we removed track changes that show all our edits.
Comment #1: Abstract Lines 14 – 16: This statement is an incorrect interpretation of the results in Table 2.
Response #1: Thank you for your comment. The statements have been revised (page 1, line 16 ~ 20).
Comment #2: Introduction Lines 24 – 28: The incorrect citation is given here. See doi: 10.1016/j.amepre.2021.05.035
Response #2: Thank you for your correction. Reference have been revised.
Comment #3: Materials and Methods Survey data analysis procedures were not described. I would anticipate that the DLDP applies sampling weights. If so, the data analyses and tables need to be corrected.
Response #3: Thank you for your comments. According to comments, data have been re-analyzed using sampling weights (page 3, line 109; Table 1 & Table 2).
Comment #4: Results Table 1: Please adjust the column label for the “No suicide ideation” group to align with the “n” column for that group.
Response #4: Thank you for your comment. It has been corrected (Table 1).
Comment #5: Lines 137 – 138: The outcome variable for suicidal ideation was binary, while this statement is expressed as a count variable. Please correct or clarify.
Response #4: Thank you for your comment. It has been corrected (pave 5, line 170 ~ line 172).
Comment #6: Discussion Line 149: The percentage of PWPD with suicidal ideation (18.7%) is not easily discernable from Table 1, nor is it given in the Results section. Please make adjustments to make this clearer.
Response #6: Thank you for your valuable comment. I have inserted the results in Table 1.
Comment #7: Suggest primarily focusing on the adjusted results from Table 2 for the Discussion section.
Response #7: Based on your comments, discussion have been revised for focusing on the adjusted results (pave 5, line 188 line 194; page 6, line 216 ~ line 225).
Comment #8: Lines 170 – 173: Like the abstract, this statement is an incorrect interpretation of the results in Table 2.
Response #8: Thank you for your comment. It has been corrected (pave 6, line 202 ~ line 204).
Comment #9: Conclusions
Lines 212 – 214: As previously described, this statement is an incorrect interpretation of the results in Table 2.
Response #9: Thank you for your comment. It has been corrected (pave 7, line 280 ~ line 283).
Comment #10: Comments on the Quality of English Language In general, the paper could use significant editing to reduce redundancy in the text.
Response #11: Editing have been completed by English professional.